# Synthetic Antimicrobial Immunomodulatory Peptides: Ongoing Studies and Clinical Trials

**DOI:** 10.3390/antibiotics11081062

**Published:** 2022-08-05

**Authors:** Małgorzata Lesiuk, Małgorzata Paduszyńska, Katarzyna E. Greber

**Affiliations:** 1Polygen Sp. z o.o., Portowa 16L/130, 44-100 Gliwice, Poland; 2Department of Inorganic Chemistry, Faculty of Pharmacy, Medical University of Gdansk, Al. Gen. J. Hallera 107, 80-416 Gdansk, Poland; 3Department of Physical Chemistry, Faculty of Pharmacy, Medical University of Gdansk, Al. Gen. J. Hallera 107, 80-416 Gdansk, Poland

**Keywords:** antimicrobial peptides, immunomodulatory peptides, antibiotic resistance, clinical trials

## Abstract

The increasingly widespread antimicrobial resistance forces the search for new antimicrobial substances capable of fighting infection. Antimicrobial peptides (AMPs) and their synthetic analogs form an extensive group of compounds of great structural diversity and multifunctionality, different modes of antimicrobial action, and considerable market potential. Some AMPs, in addition to their proven antibacterial, antifungal, and antiviral activity, also demonstrate anti-inflammatory and immunomodulatory capabilities; these are called innate defense regulator (IDR) peptides. IDR peptides stimulate or inhibit the body’s immune system, e.g., by stimulating leukocyte migration to the site of infection, driving macrophage differentiation and activation, providing chemotactic action for neutrophils, degranulation and activation of mast cells, altering chemokine and cytokine production, and even induction of angiogenesis and wound healing. Such multifunctional immunomodulatory peptide molecules are currently being investigated and developed. Exploring and utilizing IDR peptides as an indirect weapon against infectious diseases could represent a completely new strategy to cope with the issue of antimicrobial resistance.

## 1. Introduction

The combating of microbial infections is a constant arms race involving the invention of new types of weapons or the improvement of the existing arsenal. It is estimated that resistance to customary antibiotics is currently responsible for 700,000 deaths per year worldwide. The far-reaching forecasts, if no action is taken and the current rate of AMR development is maintained, are not optimistic, envisaging that 10 million people would be killed annually by 2050 [1]. The economic costs of decreased global production are projected to reach USD 100 trillion by 2050. The low investment outlays of the private and public sectors in searching for new antibiotics are not helping to solve the problem of serious infections that do not respond to available antibiotics. The faster we react by increasing investments in research on new antimicrobial substances, the smaller the costs that will be incurred in the future. One promising group of compounds that can be explored and exploited is antimicrobial peptides (AMPs) and their synthetic analogs. AMPs are compounds of great structural diversity, different modes of antimicrobial action, and considerable market potential. In addition to their direct antimicrobial activity, AMPs reveal anti-inflammatory and immunomodulatory properties. Innate defense regulator (IDR) peptides can stimulate or inhibit the body’s immune system, e.g., by stimulating leukocyte migration to the site of infection, driving macrophage differentiation and their activation, providing chemotactic action for neutrophils, degranulation and activation of mast cells, altering chemokine and cytokine production, and even induction of angiogenesis and wound healing. Such multifunctional immunomodulatory peptide molecules are currently being investigated and developed. Exploiting them as an indirect weapon against infectious diseases could represent a completely new strategy to cope with the issue of antimicrobial resistance [2,3].

## 2. Peptides Derived from Naturally Occurring Molecules

Many studied immunomodulatory peptides are derivatives or analogs of naturally occurring antimicrobial peptides or proteins.

### 2.1. cNK-2

cNK-2 (RRQRSICKQLLKKLRQQLSDALQNNDD) is a core α-helical region (aa 39–65) of chicken NK-lysin (cNK-lysin). It was found that cNK-2 can kill *Eimeria* sporozoites by disrupting the parasitic membrane, and its antimicrobial activity is higher than that of the original cNK-lysin peptide. It was also observed that cNK-2 administered in ovo and intraperitoneally to *E. acervulina*-infected chickens exerted a protective effect against *Eimeria* infection, and resulted in increased body weight and reduced gut lesion scores. Kim et al. studied the immunomodulatory properties of cNK-2, and found that the peptide upregulated the expression of CCL4 (5.3-fold), CCL5 (5.8-fold), and interleukin (IL)-1β in the chicken macrophage cell line HD11, and significantly upregulated CCL4 (22.5-fold) and CCL5 (13.2-fold) in primary monocytes. The peak of chemokines and cytokines was observed at 4 h after stimulation by cNK-2 in both cell types. The upregulation of chemokines and cytokines by cNK-2 was dose-dependent. Further studies by Kim et al. revealed that cNK-2 is able to suppress the expression of the pro-inflammatory cytokine IL-1β induced by LPS in HD11 cells and in primary monocytes, by 71.1% and 83.2% respectively. Additionally, cNK-2 reduced the production of NO induced by LPS in HD11 cells and monocytes, but this change was not significant. The chemokine induction activity of cNK-2 involves the mitogen-activated protein-kinase-mediated signaling (MAPK) pathway, including p38, extracellular signal-regulated kinase 1/2 (ERK), and c-Jun N-terminal kinases (JNK), as well as the internalization of cNK-2 into the cells [4].

### 2.2. Bac2A

Bac2A (RLARIVVIRVAR-NH_2_) is a linear derivative of bactenecin where two cysteines have been replaced with two alanines. It presents modest broad-spectrum antibacterial activity. MICs were noted between 2 and 32 μg/mL for Gram-negative bacteria, and between 0.25 and 16 μg/mL for Gram-positive bacteria—similar to the activity of bactenecin. Bowdish et al. studied its biological properties and established that Bac2A presents a very weak anti-endotoxic effect—about 30% at the concentration of 20 μg/mL—despite having a strong binding affinity for LPS, and was unable to block LPS-induced tumor necrosis factor alpha (TNF-α) production at the same concentration. However, Bac2A at a concentration ≥ 10 μg/mL induced chemotaxis of undifferentiated THP-1 cells [5].

### 2.3. Clavanin-MO

Clavanin-MO (FLPIIVFQFLGKIIHHVGNFVHGFSHVF-NH_2_) is the analog of clavanin A—a naturally occurring AMP (VFQFLGKIIHHVGNFVHGFSHVF-NH_2_) isolated from the hemocytes of marine tunicates. Silva et al., in an effort to improve the antimicrobial activity and immunomodulatory properties of clavanin A, introduced the short hydrophobic peptide fragment FLPII—present in a number of different immunomodulatory peptides—to the native sequence. As a result, clavanin-MO presented increased antimicrobial activity compared to the native peptide (Table 1), and revealed immunomodulatory activities in vitro and in vivo. Studies performed on murine macrophage-like cells stimulated with 10 ng/mL LPS revealed the anti-inflammatory activities of clavanin A and clavanin-MO at the concentration of 2 μM. Both peptides significantly increased the production of IL-10 and reduced the expression of pro-inflammatory IL-12 and TNF-α. In vivo studies indicated that clavanin-MO is able to induce chemotaxis. Silva et al. noted that the peptides stimulated the migration of leukocytes, and observed a significant increase in their number in the peritoneal fluid of healthy mice. On the other hand, clavanin-MO is able to increase the expression of GM-CSF, IFN-γ, and MCP-1 in the early infection phase of *E. coli* and *S. aureus* in mice. Toxicity studies were performed on murine red blood cells, murine macrophage-like cells, murine fibroblast cells, and human embryonic kidney cells. As a result, clavanin-MO showed no hemolytic activity and no cytotoxic effect on the tested cell lines at the concentrations needed for antimicrobial and immunomodulatory activity. Moreover, the peptide did not cause any apparent toxicity in mice treated intraperitoneally with a dose of 50 mg/kg [6].

### 2.4. Ac-ApoE (133–149)-NH_2_

ApoE, a large glycosylated protein, is responsible for the transport of lipids through the blood and central nervous system. A specific region of this protein, located between amino acid residues 130 and 162, is described as being crucial for its biological activity. Many studies have been conducted on this fragment and its various synthetic variants. Azuma et al. showed that a peptide with an amino acid sequence corresponding to residues 133–162 of the ApoE protein presented broad-spectrum antimicrobial activity. Based on this fragment, Pane et al. obtained four ApoE-derived peptides: rApoE_PM_ (133–150)—a recombinant ApoE peptide (133–150) with additional two residues (PM) at the N-terminus expressed in *E. coli*; sApoE (133–150)—a synthetic peptide fragment; Ac-ApoE (133–150)-NH_2_—a synthetic peptide fragment with an acylated N-terminus and amidated C-terminus; and Ac-ApoE (133–149)-NH_2_. All of these analogs have been subjected to antimicrobial tests and showed identical high activity against Gram-negative strains, along with varied activity against Gram-positive strains (Table 2). 

Tests performed on PMA-differentiated THP-1 (THP-1D) stimulated with LPS indicated that rApoE PM (133–150) decreases the expression of IL-8 and COX-2. Similar activity has been observed for sApoE (133–150), COG-133, and Ac-ApoE (133–150)-NH_2_. On the other hand, none of the tested peptides significantly influenced the expression of cytokines in undifferentiated THP-1 cells (THP-1U). Treatment of LPS-stimulated peripheral blood mononuclear cells (PBMCs) with sApoE (133–150) caused a significant reduction in the amount of tumor necrosis factor alpha (TNF-α) released, from 1478.38pg/mL to 683.63 pg/mL. Other studies showed that sApoE (133–150) is able to induce the expression of monocyte chemotactic protein-1 (MCP-1)—a cytokine involved in leukocyte migration—leading to the assumption that the tested peptide can play a role in promoting migration and recruitment of monocytes/macrophages in response to inflammation and tissue injury. It was also confirmed that rApoE PM (133–150) decreases the expression of COX-2 and IL-8 in LPS treated HaCaT cells [7].

## 3. De Novo Designed Antimicrobial Immunomodulatory Peptides

### 3.1. KSLW

KSL (KKVVFKVKFK-NH2) is a peptide developed by combinatorial chemistry and designed to be active against *Candida albicans*. It has an exceptional amino acid sequence with five basic residues of lysine, giving a net positive charge of +6. The minimum inhibitory concentration (MIC) and minimum fungicidal concentration (MFC) of KSL for *C. albicans* was determined at the level of 0.78 μg/mL. This peptide was also identified as having activity against Gram-positive and Gram-negative bacteria. The MIC of KSL for *Staphylococcus aureus* and *Pseudomonas aeruginosa* was 0.78 μg/mL and 1.56 μg/mL, respectively. KSLW (KKVVFWVKFK-NH_2_), which is derived from KSL, maintained the broad spectrum of antimicrobial activity, but also gained immunomodulatory properties. Williams et al. assessed the chemotactic effect of KSLW on human neutrophils. Their studies revealed that KSLW displays chemoattractant activity for neutrophils, as they migrated toward the peptide in a concentration-dependent manner, ranging from 10^−7^ M to 10^−3^ M, with the highest level of migration (>90%) at 10^−5^ M. It was also indicated that KSLW is able to induce actin polymerization in neutrophils and exert an anti-inflammatory effect on phorbol myristate acetate (PMA)- and LPS-stimulated neutrophils [8]. Lee et al. developed KSLW analogs modified with polyethylene glycol (PEG)-aldehyde and distearoylphosphatidylethanolamine (DSPE)-PEG-aldehyde, obtaining PEG–KSLW and PLM–KSLW, respectively. These two PEGylated peptide derivatives were tested for their antibacterial and antiseptic effects in vivo. Studies were performed on a murine model of cecal ligation and puncture (CLP)-induced sepsis, and revealed that the treatment of septic mice with KSLW, PEG–KSLW, and PLM–KSLW increased the survival rate of mice compared to untreated ones. Additionally, treatment with KSLW and its derivatives decreased the mortality in septic mice intravenously injected with LPS, as well as those infected with Gram-negative bacteria [9].

### 3.2. SET-M33

SET-M33 is a peptide developed by research group from Siena. It consists of four identical short peptides linked to the branched structure by a lysine core (Figure 1). Due to its strong activity against Gram-negative bacteria, it is currently under preclinical study as a new drug candidate to treat lung infections.

In addition to antimicrobial activity, SET-M33 was also proven to present strong anti-inflammatory activity. Brunetti et al. investigated the ability of SET-M33 to neutralize LPS and inhibit the inflammatory cytokines TNF-α, IL1-β, MIP1, MIP2, IL-6, GM-CSF, KC, and IP10. Studies revealed that murine macrophages stimulated with LPS from *K. pneumoniae*, *P. aeruginosa*, or *E. coli* and treated with SET-M33 expressed the pro-inflammatory cytokines in a significantly reduced amount, i.e., 100% for KC in macrophages stimulated with LPS from *K. pneumoniae*, and for KC and IP10 in macrophages stimulated with LPS from *E. coli*; >90% for IL-6 in cells stimulated with LPS from all bacteria; >90% for GM-CSF in cells incubated with LPS from *K. pneumoniae*; >90% for IP10 in cells stimulated with LPS from *P. aeruginosa*; >90% for MPC1 in cells incubated with LPS from *E. coli*; >75% for IL1-β, MIP1, and MCP1 in cells stimulated with LPS from *P. aeruginosa*; and >75% for TNF-α, MIP1, and GM-CSF in cells stimulated with LPS from *E. coli*. It was also confirmed that SET-M33 is able to effectively decrease the expression of iNOS—the enzyme crucial for the generation of nitric oxide—and COX-2—the enzyme responsible for production of prostanoids—with >75% efficiency in cells stimulated with LPS from *K. pneumoniae* and *P. aeruginosa*. The tested peptide completely inhibited the expression of NF-κB in RAW264.7 macrophages stimulated with 25 ng/mL LPS from *K. pneumoniae*. SET-M33 presented inhibitory effects on inflammatory cytokine expression in IB3–1 bronchial cells isolated from a cystic fibrosis patient. The following inhibition of cytokine expression in the cells incubated with LPS was observed: >20% IL-6 and IL-8; >14% G-CSF; >12% VEGF. LPS neutralization studies performed in vivo on an animal model showed that mice treated with LPS and SET-M33, at a dose of 5 mg/kg, produced 99% less TNF-α than animals treated with LPS only. SET-M33 also presented the ability to promote cell migration. An in vitro assay, using keratinocytes (HaCaT), confirmed that SET-M33 promotes the closure of the 1 μm gap in the cell monolayer by promoting cell motility [10].

### 3.3. GW-A2

Chou et al., using a template-assisted approach [11], developed the peptide structure GAKYAKIIYNYLKKIANALW, with high antibacterial potential. This peptide, named GW-A2, presented considerable activity toward Gram-positive bacteria and weak activity against Gram-negative bacteria. 

Further studies by Li et al. proved the anti-inflammatory activity of GW-A2, which inhibited the generation of NO in *E*. *coli*-LPS- and *Salmonella*-LPS-activated macrophages. Additionally, GW-A2 inhibited the expression of iNOS and COX-2, and reduced the secretion levels of TNF-α and IL-6, in *E*. *coli*-LPS-activated macrophages. Surprisingly, GW-A2 at a concentration of 2 μM reduced the COX-2 expression more effectively than at 4 μM. GW-A2 significantly reduced the generation of LPS-induced NO in RAW 264.7 cells when it was added to the cells either with LPS or 24 h before adding LPS. Furthermore, GW-A2 added to RAW 264.7 cell culture at 0–8 h after LPS stimulation significantly inhibited NO generation, but it had no effect when it was added 16 h after LPS stimulation. It was observed that in intraperitoneally LPS-injected mice, GW-A2 significantly reduced the levels of IL-1β, IL-6, and TNF-α in the sera, and reduced the expression of iNOS, COX-2, and NLRP3 in the lungs and liver, indicating that the tested peptide has anti-inflammatory properties in vivo [12].

### 3.4. AWRK6

Dybowskin-2CDYa (SAVGRHSRRFGLRKHRKH), the antibacterial peptide naturally present in the skin of the *Rana dybowskii* frog, was used as a template to design the more hydrophobic and trypsin-resistant AWRK6 peptide (SWVGKHGKKFGLKKHKKH). Replacement of the Arg residues originally present in dybowskin-2CDYa with Lys residues resulted in higher affinity to anionic liposomes than to zwitterionic ones. Studies of antimicrobial activity indicated that AWRK6 was more active against Gram-positive than Gram-negative bacteria, and was 2–4-fold more potent compared to dybowskin-2CDYa, being found to be at the level of 4–8 μg/mL against *S. aureus* and 32 μg/mL against *E. coli* [13].

Wang et al., using an enzyme-linked immunosorbent assay (ELISA), examined the efficacy of AWRK6 in terms of LPS neutralization, and revealed that the peptide reduced the binding of LBP with LPS by up to 70%, in a dose- and time-dependent manner. Moreover, the LAL assay indicated that AWRK6 binds to LPS and, therefore, can inhibit the inflammatory response [14]. 

Selected peptides and peptide derivatives with proven immunomodulatory properties are summarized in Table 3.

### 3.5. Synthetic Antimicrobial Immunomodulatory Peptides under Clinical Trials

There are many AMPs in clinical trials—both synthetic AMPs and those of natural origin. The synthetic peptides with immunomodulatory effects include the following:

Brilacidin (PMX-30063) is a synthetic, immunomodulatory AMP, successfully tested in phase II clinical trials for the treatment of acute bacterial skin and skin structure infections. In addition, it has been proven to have antiviral activity, inhibiting the SARS-CoV-2 virus in Vero African green monkey kidney cells and Calu-3 human lung epithelial cells, and showing a synergistic inhibitory activity in combination with the antiviral remdesivir [15].

PXL01 is an analog of human lactoferrin, shown in phase II clinical trials to be a safe, well tolerated, and effective postoperative anti-adhesion treatment after tendon repair surgery in an inpatient [16].

Omiganan (MBI-226) is a derivative of indolicidin that has been tested in a total of 16 studies, 13 of which have been completed. Looking at the completed trials, three phase III studies have been reported, among which two were aimed at evaluating the efficacy of AMPs as topical gel formulations to treat rosacea. The third phase III study concerned the treatment of catheter colonization, along with the prevention of bloodstream infections if applied to the skin surrounding the insertion [15].

Pexiganan (MSI-78) is analog of magainin, isolated from the African clawed frog Xenopus laevis, having ended phase III clinical trials as a topical cream for the treatment of infected diabetic foot ulcers. Because the results were not better than those obtained with ofloxacin, further clinical trials were discontinued [16].

IDR-1 (bactenecin) is a synthetic peptide derived from natural CHDPs (cationic host defense peptides). In phase I clinical trials, it exhibited extensive antimicrobial activity against bacteria—particularly Gram-negative bacteria—along with strong cytotoxicity. IDR-1 has been shown to control inflammation in various animal models of infection and sepsis [16].

**Table 3 antibiotics-11-01062-t003:** Selected peptides and peptide derivatives with proven immunomodulatory properties.

De Novo Designed Synthetic Peptides			
Peptide	Sequence	Activity	References
**KSLW**	KKVVFWVKFK-NH_2_	Chemoattractant activity for neutrophils; induces actin polymerization in neutrophils; exerts an anti-inflammatory effect on phorbol myristate acetate (PMA)- and LPS-stimulated neutrophils.	[17]
**mAb HuA V** ** _L_ ** **CDR3**	GQTTVTKIDEDY	Induces a significant upregulation of IL-6 production.	[18]
**mAb MoA V** ** _H_ ** **CDR3**	GQYGNLWFAY	Induces an increased production of both IL-6 and TNF-α.	[18]
**SET-M33**	KKIRVRLSA	Inhibition of the inflammatory cytokines TNF-α, IL1-β, MIP1, MIP2, IL-6, GM-CSF, KC, and IP10; decreases the expression of iNOS and COX-2; inhibitory effects on inflammatory cytokine expression in IB3–1 bronchial cells; in vivo neutralization of LPS; promotes keratinocyte migration.	[10]
**GW-A2**	GAKYAKIIYNYLKKIANALW	Inhibition of the expression levels of nitric oxide (NO), inducible NO synthase (iNOS), cyclooxygenase-2 (COX-2), tumor necrosis factor-α (TNF-α), and interleukin-6 (IL-6) in lipopolysaccharide (LPS)-activated macrophages.	[11,12]
**Dusquetide** **(SGX942)**	RIVPA	Modulates the innate immune system at key convergence points in intracellular signaling pathways, and demonstrates activity in both reducing inflammation and increasing clearance of bacterial infection.	[19]
**Pep19-2.5** **Pep19-4LF**	GCKKYRRFRWKFKGKFWFWG GKKYRRFRWKFKGKLFLFG	Anti-inflammatory activity of Pep19-2.5 is associated not only with neutralization of cell-free bacterial toxins, but also with a direct binding of the peptide to the outer leaflet of the bacterial outer membrane.	[20]
**Synthetic Peptides Based on Naturally Occurring Molecules**			
**Peptide**	**Sequence**	**Activity**	**References**
**cNK-2**	RRQRSICKQLLKKLRQQLSDALQNNDD	Upregulates the expression of CCL4, CCL5, and interleukin (IL)-1β in chicken macrophages; suppresses the expression of the pro-inflammatory cytokine IL-1β induced by LPS in HD11 cells and in primary monocytes; reduces the production of NO induced by LPS in HD11 cells and monocytes.	[4]
**Bac2a**	RLARIVVIRVAR-NH_2_	Anti-endotoxic activity; induces chemotaxis of undifferentiated THP-1 cells.	[21]
**clavanin-MO**	FLPIIVFQFLGKIIHHVGNFVHGFSHVF-NH_2_	Increases the production of IL-10 and reduces the expression of pro-inflammatory IL-12 and TNF-α; stimulates the migration of leukocytes; increases the expression of GM-CSF, IFN-γ, and MCP-1 in the early infection phase.	[6]
**Ac-ApoE (133–149)-NH_2_**	Ac-LRVRLASHLRKLRKRLL-NH_2_	Anti-inflammatory/neuroprotective activity and antimicrobial activity; binding of lipoprotein receptor-related protein (LRP) by apoE (133–149) results in inhibition of the N-methyl-D-aspartate (NMDA) receptor (NMDAR).	[7]

Many studies have shown that the activity of AMPs is mostly associated with bacterial membrane destabilization [22,23,24,25]; however, their activity expressed as MIC is often much lower than that of conventional antibiotics [26,27]. Exceptional cases, when AMPs present higher antimicrobial potency (i.e., lower MIC values) than conventional antibiotics, arise when germs have developed resistance to the latter [6]. For that reason, if AMPs are to one day be an active pharmaceutical ingredient (API) in anti-infective drug formulation, it seems necessary to equip the AMP molecule with more than just direct antimicrobial activity. The promising future development path of AMPs as new therapeutics is to create multifunctional AMPs by combining their direct antimicrobial activity with indirect activity, e.g., modulation of the immune system response. Immunomodulation can enhance the antimicrobial effects of AMPs by inducing the recruitment of antigen-presenting cells to the site of infection, activating neutrophils or macrophages, influencing the differentiation of dendritic cells, and suppressing pro-inflammatory cytokines [16,28,29,30]. Anti-infective therapeutic peptides with combined antimicrobial and immunomodulatory properties represent a new approach to treat antibiotic-resistant infections.

## 4. Conclusions

The current problem with the emergence of multidrug-resistant bacteria has forced scientists to look for new, alternative solutions. There are many challenges—e.g., production costs, the problem of bioavailability, the cytotoxicity of newly synthesized peptides—but the effects are promising. For example, their broad spectrum of activity, with a multitarget, non-specific, and rapid mode of action, results in limited emergence of resistance, immunomodulatory properties, and synergistic interactions with conventional antibiotics, with the potential to eliminate the threat of MDR bacteria [31].

## Figures and Tables

**Figure 1 antibiotics-11-01062-f001:**
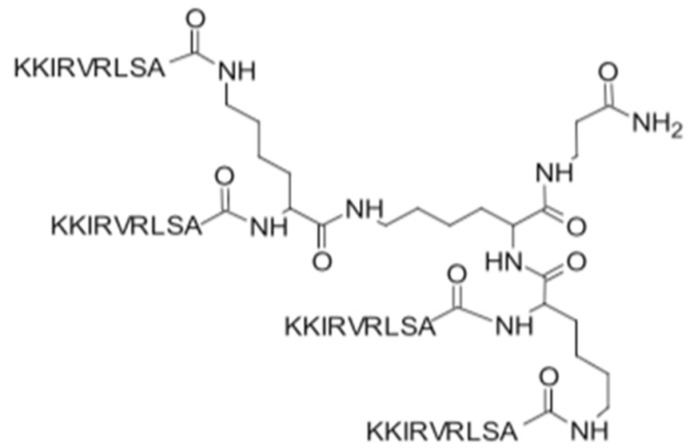
Structure of the SET-M33 peptide [10].

**Table 1 antibiotics-11-01062-t001:** Antibacterial activity of clavanin A and the synthetic peptide clavanin-MO. Adapted from ref. [6].

Microorganism	MIC (μM)	
	Clavanin A	Clavanin-MO
*Bacillus subtilis* ATCC6633	6.0	1.5
*Enterococcus faecalis* ATCC12953	6.0	1.5
*Staphylococcus aureus* ATCC29213	48.0	24.0
*Staphylococcus aureus* (MRSA) ATCC33591	12.0	6.0
*Escherichia coli* ATCC8739	24.0	12.0
*Escherichia coli* KPC-positive (1812446)	48.0	6.0
*Klebsiella pneumoniae* ATCC13885	6.0	3.0
*Klebsiella pneumoniae* 1825971 (KPC971)	6.0	3.0
*Pseudomonas aeruginosa* ATCC 15442	12.0	3.0

**Table 2 antibiotics-11-01062-t002:** Antibacterial activity of ApoE-derived peptides. Adapted from ref. [7].

Microorganism	MIC (μM)			
	rApoE _PM_ (133–150)	sApoE (133–150)	Ac-ApoE (133–150)-NH_2_	Cog-133
*Escherichia coli* DH5α	12.5	12.5	12.5	12.5
*E. coli* ATCC 25922	6.25	6.25	6.25	6.25
*P. aeruginosa* PAO1	25	25	25	25
*P. aeruginosa* PA14	50	50	50	50
*K. pneumoniae* ATCC 700603	6.25	6.25	3.12	3.12
*S. aureus* ATCC 6538P	3.12	3.12	100	100
*B. subtilis* PY79	25	25	>100	>100

## Data Availability

Not applicable.

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
