# Peer review of "Synthetic Antimicrobial Immunomodulatory Peptides: Ongoing Studies and Clinical Trials"

_antibiotics, 2022, doi:10.3390/antibiotics11081062_

Round 1

Reviewer 1 Report

In their manuscript entitled "Synthetic antimicrobial immunomodulatory peptides- ongoing studies and clinical trials report.", prof K.E. Greber introduced the recent studies on the antimicrobial peptides as immunomodulatory reagents. Their manuscript covered the recent application of antimicrobial peptides in clinical trials. However, a substantial revision is needed to make this manuscript for publication in antibiotics journal.

My major comments are as follows.

1) The reference number was wrong. The author should fix them.

For example, ref. 5 in the manuscript by D.M. Bowdish are described as ref 12 in reference section. Moreover, where is the ref 11-13 in the manuscript?

2) In page 2 line 2 from bottom, the Table X should be Table 1.

3) In Table 2, the name of microorganism should be italic.

4) The authors should add the sequence of all antimicrobial peptides. For example, the sequence of cNK2, Bac2A, and Ac-ApoE , were lacked.

Author Response

Dear Reviewer, 

We are very gratefull for your time and comments on the manuscript.

Please find below our answers:

1) The reference number was wrong. The author should fix them.

For example, ref. 5 in the manuscript by D.M. Bowdish are described as ref 12 in reference section. Moreover, where is the ref 11-13 in the manuscript?

References are fixed. Thank you.

2) In page 2 line 2 from bottom, the Table X should be Table 1.

Number of Table has been corrected

3) In Table 2, the name of microorganism should be italic.

Names of the microorganisms are in italic now.

4) The authors should add the sequence of all antimicrobial peptides. For example, the sequence of cNK2, Bac2A, and Ac-ApoE , were lacked.

Sequences have been added

Reviewer 2 Report

the authors present a review concerning antimicrobial peptides, also showing anti-inflammatory activity and immunomodulating capabilities, called innate defense regulator (IDR) peptides.

4 natural peptides and 4 synthetic peptides are described in the review. Their antibacterial and immunomodulatory properties are described. While the antimicrobial properties of peptides are well known, IRD/AMP combined peptides are less so. The review is therefore of definite interest. However, two remarks can be made:

-the choice of 8 peptides: knowing the very large number of AMPs, (Biopolymers. 2013 Nov; 100(6): 572–583, Mar Drugs. 2019 Jun; 17(6): 350) the authors could explain the reasons which led to the selection of these 8 peptides

-Perhaps the authors could have been interested in the mechanisms of action of these peptides in order to have a less descriptive publication.

A final remark concerns the peptide sequences to be included in the publication. The molecular aspect should not be ignored.

Author Response

Dear Reviewer,

We are very grateful for your time and kind comments. The manuscript has been corrected to the comments given by all reviewers. We hope its current quality will be sufficient. Please find our answers to tour comments below:

-the choice of 8 peptides: knowing the very large number of AMPs, (Biopolymers. 2013 Nov; 100(6): 572–583, Mar Drugs. 2019 Jun; 17(6): 350) the authors could explain the reasons which led to the selection of these 8 peptides

It is true that there is a large number of AMPs described in the literature, however, our goal was to describe synthetic peptides (those of de novo designed and those based on naturally occurring structures) with both antimicrobial and immunomodulatory properties, and if you put it this that way, there is a narrow selection of structures that meet the assumptions. 

-Perhaps the authors could have been interested in the mechanisms of action of these peptides in order to have a less descriptive publication.

Of course, the mechanism of action is very important part of antimicrobial peptides issue. There is number of papers which describe it widely. In our manuscript we intended to describe in details not only the direct antimicrobial activity but also the indirect one. This approach required full description of these properties and not only general mechanism. I hope our manuscript can be accepted by the Reviewer as it is.  

A final remark concerns the peptide sequences to be included in the publication. The molecular aspect should not be ignored.

Sequences of described peptides have been added

Reviewer 3 Report

I am wondering if the author could write a table summarizing/classifying the immunomodulatory activities of AMPs discussed in the main text. 

I am wondering if the author could write another separate paragraph before the conclusion section to comment/discuss about the future direction of the development of e.g. multifunctional antimicrobial peptides?

Author Response

Dear Reviewer,

We are very grateful for your time and comments on the manuscript. Please find the answers below. We hope the quality of the manuscript will satisfy you. 

Kind regards

  • I am wondering if the author could write a table summarizing/classifying the immunomodulatory activities of AMPs discussed in the main text.

Table has been added to the body text of the manuscript. 

  • I am wondering if the author could write another separate paragraph before the conclusion section to comment/discuss about the future direction of the development of e.g. multifunctional antimicrobial peptides?

The paragraph “Future directions” has been added to the manuscript.

Round 2

Reviewer 1 Report

Their revised version of manuscript were totally improved. All revised points that I pointed out were corrected. I recommend the manuscript to publish for Antibiotics journal as review.